# CONFIDENCE-AWARE TRAINING OF SMOOTHED CLASSIFIERS FOR CERTIFIED ROBUSTNESS

## ABSTRACT

Any classifier can be "smoothed out" under Gaussian noise to build a new classifier that is provably robust to $\ell_2$-adversarial perturbations, *viz.*, by averaging its predictions over the noise, namely via *randomized smoothing*. Under the *smoothed classifiers*, the fundamental trade-off between accuracy and (adversarial) robustness has been well evidenced in the literature: *i.e.*, increasing the robustness of a classifier for an input can be at the expense of decreased accuracy for some other inputs. In this paper, we propose a simple training method leveraging this trade-off for obtaining more robust smoothed classifiers, in particular, through a *sample-wise* control of robustness over the training samples. We enable this control feasible by investigating the correspondence between robustness and *prediction confidence* of smoothed classifiers: specifically, we propose to use the "accuracy under Gaussian noise" as an easy-to-compute proxy of adversarial robustness for each input. We differentiate the training objective depending on this proxy to filter out samples that are unlikely to benefit from the worst-case (adversarial) objective. Our experiments following the standard benchmarks consistently show that the proposed method, despite its simplicity, exhibits improved certified robustness upon existing state-of-the-art training methods.

## 1 INTRODUCTION

Despite these tremendous advances in *deep neural networks* for a variety of tasks towards artificial intelligence, *e.g.*, visual recognition (He et al., 2016; Chen et al., 2020), natural language processing (Vaswani et al., 2017; Brown et al., 2020), and reinforcement learning (Silver et al., 2017; Vinyals et al., 2019), the broad existence of *adversarial examples* (Szegedy et al., 2014) is still one of the most significant aspects that reveals the gap between machine learning systems and humans: for a given input $x$ (*e.g.*, an image) to a classifier $f$, say a neural network, $f$ often permits a perturbation $\delta$ that completely flips the prediction $f(x + \delta)$, while $\delta$ is too small to change the semantic in $x$. In response to this vulnerability, there have been significant efforts in building *robust* neural network based classifiers against adversarial examples, either in forms of *empirical defenses* (Athalye et al., 2018; Carlini et al., 2019; Tramer et al., 2020), which are largely based on *adversarial training* (Madry et al., 2018; Zhang et al., 2019; Wang et al., 2020; Zhang et al., 2020b; Wu et al., 2020), or *certified defenses* (Wong & Kolter, 2018; Xiao et al., 2019; Cohen et al., 2019; Zhang et al., 2020a), depending on whether the robustness claim can be theoretically guaranteed or not.

*Randomized smoothing* (Lecuyer et al., 2019; Cohen et al., 2019), our focus in this paper, is currently a prominent approach in the context of certified defense, thanks to its scalability to arbitrary neural network architectures while previous methods have been mostly limited in network sizes or require strong assumptions on their architectures: specifically, for a given classifier $f$, it constructs a new classifier $\hat{f}$, where $\hat{f}(x)$ is defined to be the class that $f(x + \delta)$ outputs most likely over $\delta \sim \mathcal{N}(0, \sigma^2 I)$, *i.e.*, the Gaussian noise. Then, it is shown by Lecuyer et al. (2019) that $\hat{f}$ is certifiably robust in $\ell_2$-norm, and Cohen et al. (2019) further tightened the $\ell_2$-robustness guarantee which is currently considered as the state-of-the-art in certified defense.

However, even with recent methods for adversarial defense, including randomized smoothing, the *trade-off* between robustness and accuracy (Tsipras et al., 2019; Zhang et al., 2019) has been well evidenced, *i.e.*, increasing the robustness for a specific input can be at the expense of decreased accuracy for other inputs. For instance, with the current best practices, Salman et al. (2020) report

that the accuracy of ResNet-50 on ImageNet degrades, *e.g.*, 75.8% → 63.9%, by an $\ell_\infty$-adversarial training, *i.e.*, optimizing the classifier to ensure robustness at all the given training samples around an $\ell_\infty$-ball (of size $\frac{4}{255}$). In addition, Zhang et al. (2019) have shown that the (empirical) robustness of a classifier can be further boosted in training by paying more expense in accuracy. A similar trend can be also observed with certified defenses, *e.g.*, randomized smoothing, as the clean accuracy of smoothed classifiers are usually less than those one can obtain from the standard training on the same architecture (Cohen et al., 2019).

**Contribution.**    In this paper, we develop a novel training method for randomized smoothing, coined *Confidence-Aware Training for Randomized Smoothing* (CAT-RS), which incorporates a *sample-wise* control of target robustness on-the-fly motivated by the accuracy-robustness trade-off in smoothed classifiers. Intuitively, a natural approach one can consider in response to the trade-off in robust training is to appropriately lower the robustness requirement for "hard-to-classify" samples while maintaining those for the remaining ("easier") samples: here, the challenges are (a) which samples we should choose for it during training and (b) how to control their target robustness.

To implement this idea, we focus on the direct correspondence from *prediction confidence* to adversarial robustness that smoothed classifiers offer: due to its local-Lipschitzness (Salman et al., 2019), achieving a high confidence at $x$ from a smoothed classifier also implies a high (certified) robustness at $x$. Inspired by this, we propose to use the sample-wise confidence (of smoothed classifiers) as an efficient proxy of the certified robustness, and defines two new losses, namely *bottom-K* and *worst-case* Gaussian training, each of those targets different levels of confidence so that the overall training can be more informed sample-wise for better robustness by preventing low-confident samples from being enforced to increase their robustness.

We verify the effectiveness of our proposed method through an extensive comparison with existing robust training methods for smoothed classifiers, including the state-of-the-arts, on a wide range of established benchmarks on MNIST and CIFAR-10 datasets. Our experimental results constantly show that the proposed method can significantly improve the previous state-of-the-art results on certified robustness achievable from a given neural network architecture, by (a) maximizing the robust radii of high-confidence samples while (b) reducing the risk of deteriorating the accuracy at low-confidence samples. Our extensive ablation study further confirms that each of both proposed components has an individual effect on improving certified robustness, and can effectively control the accuracy-robustness trade-off with the hyperparameter between the two proposed losses.

## 2 PRELIMINARIES

**Adversarial robustness.**    Consider a labeled dataset $\mathcal{D} = \{(x_i, y_i)\}_{i=1}^n$ from a certain distribution $P$, where $x \in \mathbb{R}^d$ and $y \in \mathcal{Y} := \{1, \cdots, K\}$, which forms a classification problem with $K$ classes. Let $f : \mathbb{R}^d \to \mathcal{Y}$ be a classifier. Notice that $f$ is a discrete and non-differentiable, so that one can additionally consider a differentiable $F : \mathbb{R}^d \to \Delta^{K-1}$ to allow a gradient-based optimization assuming $f(x) := \arg\max_{k \in \mathcal{Y}} F_k(x)$, where $\Delta^{K-1}$ is probability simplex in $\mathbb{R}^K$. The standard framework of *empirical risk minimization* to optimize $f$ assumes that the samples in $\mathcal{D}$ are *i.i.d.* from $P$ and expect $f$ to perform well given that the future samples also follow the *i.i.d.* assumption.

However, in the context of *adversarial robustness* (and for other notions of robustness as well), the *i.i.d.* assumption on the future samples does not hold anymore: instead, it additionally assumes that the samples can be *arbitrarily* perturbed up to a certain restriction, *e.g.*, a bounded $\ell_2$-ball, and focuses on the *worst-case* performance over the perturbed samples. One possible way to quantify this scenario is to consider the *average minimum-distance* of adversarial perturbation (Moosavi-Dezfooli et al., 2016; Carlini & Wagner, 2017; Carlini et al., 2019), namely:

$$R(f; P) := \mathbb{E}_{(x,y) \sim P} \left[ \min_{f(x') \neq y} ||x' - x||_2 \right]. \tag{1}$$

In this respect, we aim to find $f$ that maximizes $R(f; P)$ while maintaining the performance on $P$.

**Randomized smoothing.**    The essential challenge in achieving adversarial robustness in neural networks, however, stems from that directly evaluating (1) (and further optimizing it) is usually computationally infeasible, *e.g.*, under the standard practice that $F$ is modeled by a complex, high-dimensional neural network. *Randomized smoothing* (Lecuyer et al., 2019; Cohen et al., 2019)

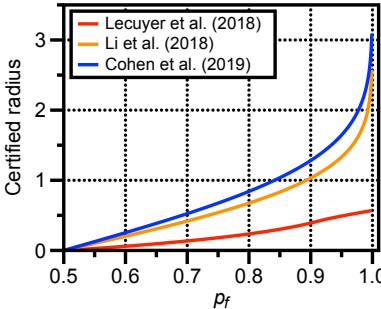
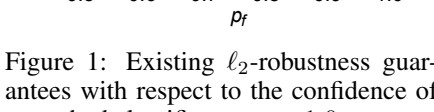
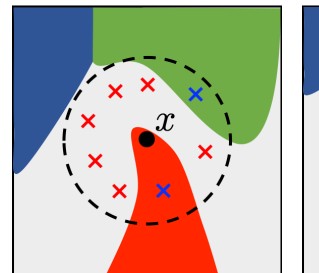
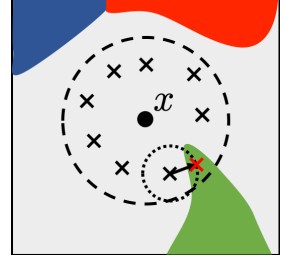

(a) Low-confidence samples (b) High-confidence samples

Figure 1: Existing $\ell_2$-robustness guarantees with respect to the confidence of smoothed classifiers at $\sigma = 1.0$.

Figure 2: Illustration of the two proposed losses, *i.e.*, *bottom-K* and *worst-case* Gaussian training, under different confidence conditions in randomized smoothing.

bypasses this difficulty by constructing a new classifier $\hat{f}$ from $f$ instead of letting $f$ to directly model the robustness: specifically, it transforms the base classifier $f$ with a certain *smoothing measure*, where in this paper we focus on the case of Gaussian distributions $\mathcal{N}(0, \sigma^2 I)$:

$$\hat{f}(x) := \arg\max_{c \in \mathcal{Y}} \mathbb{P}_{\delta \sim \mathcal{N}(0,\sigma^2 I)} \left( f(x + \delta) = c \right). \tag{2}$$

Then, the robustness of $\hat{f}$ at $(x, y)$, namely $R(\hat{f}; x, y)$, can be explicitly lower-bounded in terms of the *certified radius* $\underline{R}(\hat{f}, x, y)$, *e.g.*, Cohen et al. (2019) showed that the following bound holds which is tight for $\ell_2$-adversary, *e.g.*, it is the optimal for linear classifiers:

$$R(\hat{f}; x, y) \geq \sigma \cdot \Phi^{-1}(p_f(x, y)) =: \underline{R}(\hat{f}, x, y) \tag{3}$$

$$\text{where} \quad p_f(x, y) := \mathbb{P}_\delta(f(x + \delta) = y), \tag{4}$$

provided that $\hat{f}(x) = y$, otherwise $R(\hat{f}; x, y) := 0$.[1] Here, we remark that the formula for certified radius (3) is essentially a function of $p_f$ (4), which represents the *prediction confidence* of $\hat{f}$ at $x$, or equivalently, the *accuracy* of $f(x + \delta)$ over $\delta \sim \mathcal{N}(0, \sigma^2 I)$. In other words, unlike standard neural networks, smoothed classifiers can guarantee a correspondence from prediction confidence to adversarial robustness - which is the key motivation of our method in this paper. Figure 1 plots this relationship shown by Cohen et al. (2019) as well as by some prior works (Lecuyer et al., 2019; Li et al., 2019) which also attempt to lower-bound the robustness of smoothed classifiers.

## 3 CONFIDENCE-AWARE TRAINING FOR RANDOMIZED SMOOTHING

We aim to develop a new training method to maximize the certified robustness of $\hat{f}$, considering the trade-off relationship between robustness and accuracy (Zhang et al., 2019): even though randomized smoothing can be applied for any classifier $f$, the actual robustness of $\hat{f}$ depends how much $f$ classifies well under presence of Gaussian noise, *i.e.*, by $p_f(x, y)$ as in (3). A simple way to train $f$ for a robust $\hat{f}$, therefore, is to minimize the standard cross-entropy loss (denoted by $\mathbb{CE}$ below) with Gaussian augmentation as in Cohen et al. (2019):

$$\min_F \mathbb{E}_{\substack{(x,y) \sim P \\ \delta \sim \mathcal{N}(0,\sigma^2 I)}} \left[ \mathbb{CE}(F(x + \delta), y) \right]. \tag{5}$$

In this paper, we extend this basic form of training to incorporate a *confidence-aware* strategy to decide which noise samples $\delta \sim \mathcal{N}(0, \sigma^2 I)$ should be focused on sample-wise during training of $f$. Recall Figure 1 that plots mappings from the (smoothed) confidence $p_f$ (4) to a certified radius, *e.g.*, those derived by Cohen et al. (2019) (at the blue line). Ideally, one may wish to obtain a classifier $f$ that achieves $p_f(x, y) \approx 1$ for every $(x, y) \sim P$ to maximize its certified robustness. In practice, however, such a case is highly unlikely, and there usually exists a sample $x$ that $p_f(x, y)$ should be quite lower than 1 to maintain the discriminativity with other samples: in other words, these

---

[1]$\Phi$ denotes the cumulative distribution function of the standard normal distribution.

samples can be actually "beneficial" to be misclassified at some (hard) Gaussian noises, otherwise the classifier has to memorize the noises to correctly classify them. On the other hand, for the samples which can indeed achieve $p_f(x, y) \approx 1$, the current Gaussian training in (5) may not be able to provide enough samples of $\delta$ for $x$ throughout the training, as $p_f(x, y) \approx 1$ implies that $f(x + \delta)$ must be correctly classified for "almost every" possibility of $\delta \sim \mathcal{N}(0, \sigma^2 I)$. Also, considering that the radius certifiable at $x$ rapidly increases as $p_f(x, y) \to 1$ as shown in Figure 1, it is important for an overall robustness of $\hat{f}$ to increase $p_f(x, y)$, especially when it can be close to 1 at the end.

In these respects, we propose two different variants of Gaussian training (5) that address each of the possible cases, *i.e.*, whether (a) $p_f(x, y) < 1$ or (b) $p_f(x, y) \approx 1$, namely with (a) *bottom-K* and (b) *worst-case* Gaussian training, respectively. During training, the method first estimates $p_f(x, y)$ for each sample by simply computing their accuracy over $M$ random samples of $\delta \sim \mathcal{N}(0, \sigma^2 I)$, and applies different forms of loss depending on the value. In the following two sections, *i.e.*, Section 3.1 and 3.2, we provide the details on each of the proposed losses, and Section 3.3 describes how to combine the two losses and defines the overall training scheme.

### 3.1 BOTTOM-$K$ GAUSSIAN TRAINING: LOSS FOR LOW-CONFIDENCE SAMPLES

Consider a base classifier $f$ and a training sample $(x, y) \in \mathcal{D}$, and suppose that $p_f(x, y) = p \ll 1$, *e.g.*, $\hat{f}$ has a low-confidence at $x$. Figure 2(a) visualizes this scenario: in this case, by definition of $p_f(x, y)$ in (4), $f(x + \delta)$ would be correctly classified to $y$ only with probability $p$ over $\delta$, and this can imply either that (a) $x + \delta$ has not yet been adequately exposed to $f$ during the training, or (b) $x + \delta$ may be indeed hard to be correctly classified for some noise samples $\delta$, so that minimizing the training loss at these noises could harm the classifier. The design goal of our proposed *bottom-K* Gaussian training is to modify the standard Gaussian training (5) to reduce the optimization burden from (b) while minimally retaining its ability to cover enough noise samples during training for (a).

To this end, we first consider $M$ random *i.i.d.* samples of $\delta$, namely $\delta_1, \delta_2, \cdots, \delta_M \sim \mathcal{N}(0, \sigma^2 I)$. Then, one can notice that the random variables $\mathbb{1}[f(x + \delta_i) = y]$'s are also *i.i.d.*, each of which follows the Bernoulli distribution of probability $p$, given that $p_f(x, y) = p$. This means that, if the current $p_f(x, y)$ is the value one attempts to keep instead of further increasing it, the number of noise samples that should be correctly classified, which can be defined as $\sum_i \mathbb{1}[f(x + \delta_i) = y]$, would follow the *binomial distribution*, namely $K \sim \text{Bin}(M, p)$, and this motivates us to consider the following loss that only minimizes the $K$-*smallest* cross-entropy losses out of from $M$ samples:

$$L^{\texttt{low}} := \frac{1}{M} \sum_{i=1}^{K} \mathbb{CE}(F(x + \delta_{\pi(i)}), y), \quad \text{where} \quad K \sim \text{Bin}(M, p_f(x, y)). \tag{6}$$

Here, $\pi(i)$ denotes the noise index with the $i$-th smallest loss value in the $M$ samples.

Yet, the loss defined in (6) may not handle the *cold-start* problem on $p_f(x, y)$, *e.g.*, at the early stage of the training where $x + \delta$ has not been adequately exposed to $f$, so that $L^{\texttt{low}}$ can be minimized too early with an under-estimated $\hat{p}_f$. We found that, however, a simple trick of *clamping* $K$ with 1 can bypass the issue: *i.e.*, we always allow the "easiest" noise among the $M$ samples to be fed into $f$ throughout the training. In addition, we also found that it is still beneficial to constrain the samples that are not within $K$, *i.e.*, the "harder" ones, to ensure that they still have a certain level of confidence for the class $y$ even if they are misclassified: here, as a simple design, we propose to *truncate* the cross entropy loss with $\log p_0$ so that $F_y(x + \delta)$ can be at least $p_0$, *e.g.*, $p_0 = \frac{1}{20}$ as done in our experiments. In these respects, we re-define the loss in (6) as in the followings:

$$L^{\texttt{low}} := \frac{1}{M} \left( \sum_{i=1}^{K^+} \mathbb{CE}(F(x + \delta_{\pi(i)}), y) + \sum_{i=K^++1}^{M} \left[ \mathbb{CE}(F(x + \delta_{\pi(i)}), y) + \log p_0 \right]^+ \right), \tag{7}$$

where $K \sim \text{Bin}(M, p)$, $K^+ := \max(1, K)$, $[\cdot]^+ := \max(0, \cdot)$, and $p_0 \in (0, 1]$ is a hyperparameter. Note that the loss becomes the standard Gaussian training as $p_0 \to 1$ and returns to (6) as $p_0 \to 0$.

### 3.2 WORST-CASE GAUSSIAN TRAINING: LOSS FOR HIGH-CONFIDENCE SAMPLES

Next, we focus on the case when $p_f(x, y) \approx 1$, *i.e.*, $\hat{f}$ has a high confidence at $x$, as illustrated in Figure 2(b). In contrast to the previous scenario in Section 3.1 (and Figure 2(a)), now the major

drawback of Gaussian training (5) does not come from the *abundance* of hard noises during training, but from the *sparseness* of such noises: considering that one can only present a limited number of noise samples to $f$ throughout its training, naïvely minimizing (5) may not cover some "potentially hard" noise samples, and this would result in a significant harm in certified radius especially at the regime of $p_f(x, y) \approx 1$ as shown in Figure 1. The purpose of *worst-case* Gaussian training is to overcome this lack of samples via an *adversarial* search around each of the noise samples.

Specifically, given that we have $M$ samples of $\delta$ as in (7), namely $\delta_1, \delta_2, \cdots, \delta_M \sim \mathcal{N}(0, \sigma^2 I)$, we propose to modify (5) to find and minimize the *worst-case* noise (a) around an $\ell_2$-ball for each noise as well as (b) among the $M$ samples, instead of minimizing the average-case loss:

$$L^{\text{high}} := \max_i \max_{\|\delta_i^* - \delta_i\|_2 \leq \varepsilon} \mathbb{CE}(F(x + \delta_i^*), y). \tag{8}$$

We use the *projected gradient descent* (PGD) (Madry et al., 2018) to solve the inner maximization in (8): namely, we perform a $T$-step gradient ascent from each $\delta_i$ with step size $2 \cdot \varepsilon/T$ while projecting the perturbations to be in the $\ell_2$-ball of size $\varepsilon$. Here, although $T$ and $\varepsilon$ can be hyperparameters that affect the inner maximization, we simply fix them in our experiments by $\varepsilon = 1.0$ and $T = 4$. We also make sure that the *likelihood* of the optimized $\delta^*$ as *i.i.d.* Gaussian still remains high by simply normalizing the mean and standard deviation of $\delta^*$ to follow those of the original $\delta$.

**Comparison to SmoothAdv.** The idea of incorporating an adversarial search for the robustness of smoothed classifiers has been also considered in previous works (Salman et al., 2019; Jeong et al., 2021): *e.g.*, Salman et al. (2019) have proposed *SmoothAdv* that applies adversarial training (Madry et al., 2018) to a "soft" approximation of $\hat{f}$ given $f$ and $M$ noise samples:

$$x^* = \arg\max_{\|x'-x\|_2 \leq \epsilon} \left( -\log \left( \frac{1}{M} \sum_i F_y(x' + \delta_i) \right) \right). \tag{9}$$

Our method is different from the previous approaches in which part of the inputs is adversarially optimized: *i.e.*, we directly optimize the noise samples $\delta_i$'s instead of $x$, with no need to assume a soft relaxation of $\hat{f}$. This is due to our unique motivation of finding the worst-case Gaussian noise, and our experimental results in Section 4 further support the effectiveness of this approach.

## 3.3 OVERALL TRAINING SCHEME

Given the two losses $L^{\text{low}}$ and $L^{\text{high}}$ defined in Section 3.1 and 3.2, respectively, we now define the full objective of our proposed *Confidence-Aware Training for Randomized Smoothing* (CAT-RS). Overall, in order to differentiate how to combine the two losses per sample basis, we use the smoothed confidence $p_f(x, y)$ (4) as the guiding proxy: specifically, we aim to apply the worst-case loss of $L^{\text{high}}$ only for the samples where $p_f(x, y)$ is already high enough. In practice, however, one does not have a direct access to the value of $p_f(x, y)$ during training, and we estimate this with the $M$ noise samples[2] as done for $L^{\text{low}}$ and $L^{\text{high}}$, *i.e.*, by $\hat{p}_f(x, y) := \frac{1}{M} \sum_{i=1}^{M} \mathbb{1}[f(x + \delta_i) = y]$. Here, we set a simple condition of "$\hat{p}_f(x, y) = 1$" to activate $L^{\text{high}}$, and the final loss becomes:

$$L^{\text{CAT-RS}} := L^{\text{low}} + \lambda \cdot \mathbb{1}[\hat{p}_f(x, y) = 1] \cdot L^{\text{high}}, \tag{10}$$

where $\mathbb{1}[\cdot]$ is the indicator random variable, and $\lambda > 0$ is a hyperparameter. The complete procedure of computing our proposed CAT-RS loss can be found in Appendix A.

## 4 EXPERIMENTS

In this section, we evaluate the effectiveness of our proposed training scheme compared to existing state-of-the-art training methods for smoothed classifiers, based on two well-established benchmarks on MNIST (LeCun et al., 1998) and CIFAR-10 (Krizhevsky, 2009) extensively in compliance to the standard protocol of the previous works (Cohen et al., 2019; Zhai et al., 2020; Jeong & Shin, 2020; Jeong et al., 2021). Overall, the experiments show that our method can consistently outperform the previous best efforts to improve the average certified radius by (a) maximizing the robust radii of high-confidence samples while (b) better maintaining the accuracy at low-confidence samples. We

---

[2]In our experiments, we use $M = 4$ for our method unless otherwise noted.

Table 1: Comparison of ACR and approximate certified test accuracy (%) on MNIST. For each column, we set our result bold-faced whenever the value improves the Gaussian baseline. We mark the highest and lowest values of certified accuracy at each radius in blue and red colors, respectively.

| $\sigma$ | Methods | ACR | 0.00 | 0.25 | 0.50 | 0.75 | 1.00 | 1.25 | 1.50 | 1.75 | 2.00 | 2.25 | 2.50 |
|---|---|---|---|---|---|---|---|---|---|---|---|---|---|
| | Gaussian (Cohen et al., 2019) | 0.910 | 99.2 | 98.5 | 96.7 | 93.3 | 0.0 | 0.0 | 0.0 | 0.0 | 0.0 | 0.0 | 0.0 |
| | Stability (Li et al., 2019) | 0.914 | 99.3 | 98.6 | 97.1 | 93.8 | 0.0 | 0.0 | 0.0 | 0.0 | 0.0 | 0.0 | 0.0 |
| | SmoothAdv (Salman et al., 2019) | 0.932 | 99.4 | 99.0 | 98.2 | 96.8 | 0.0 | 0.0 | 0.0 | 0.0 | 0.0 | 0.0 | 0.0 |
| 0.25 | MACER (Zhai et al., 2020) | 0.921 | 99.3 | 98.7 | 97.5 | 94.8 | 0.0 | 0.0 | 0.0 | 0.0 | 0.0 | 0.0 | 0.0 |
| | Consistency (Jeong & Shin, 2020) | 0.928 | 99.5 | 98.9 | 98.0 | 96.0 | 0.0 | 0.0 | 0.0 | 0.0 | 0.0 | 0.0 | 0.0 |
| | SmoothMix (Jeong et al., 2021) | 0.932 | 99.4 | 99.0 | 98.2 | 96.7 | 0.0 | 0.0 | 0.0 | 0.0 | 0.0 | 0.0 | 0.0 |
| | **CAT-RS (Ours)** | **0.933** | 99.3 | 98.9 | **98.2** | **97.0** | 0.0 | 0.0 | 0.0 | 0.0 | 0.0 | 0.0 | 0.0 |
| | Gaussian (Cohen et al., 2019) | 1.557 | 99.2 | 98.3 | 96.8 | 94.3 | 89.7 | 81.9 | 67.3 | 43.6 | 0.0 | 0.0 | 0.0 |
| | Stability (Li et al., 2019) | 1.573 | 99.2 | 98.5 | 97.1 | 94.8 | 90.7 | 83.2 | 69.2 | 45.4 | 0.0 | 0.0 | 0.0 |
| | SmoothAdv (Salman et al., 2019) | 1.687 | 99.0 | 98.3 | 97.3 | 95.8 | 93.2 | 88.5 | 81.1 | 67.5 | 0.0 | 0.0 | 0.0 |
| 0.50 | MACER (Zhai et al., 2020) | 1.583 | 98.5 | 97.5 | 96.2 | 93.7 | 90.0 | 83.7 | 72.2 | 54.0 | 0.0 | 0.0 | 0.0 |
| | Consistency (Jeong & Shin, 2020) | 1.655 | 99.2 | 98.6 | 97.6 | 95.9 | 93.0 | 87.8 | 78.5 | 60.5 | 0.0 | 0.0 | 0.0 |
| | SmoothMix (Jeong et al., 2021) | 1.694 | 98.7 | 98.0 | 97.0 | 95.3 | 92.7 | 88.5 | 81.8 | 70.0 | 0.0 | 0.0 | 0.0 |
| | **CAT-RS (Ours)** | **1.699** | 98.6 | 98.0 | **97.0** | 95.4 | 92.8 | **88.9** | **82.3** | **70.9** | 0.0 | 0.0 | 0.0 |
| | Gaussian (Cohen et al., 2019) | 1.619 | 96.3 | 94.4 | 91.4 | 86.8 | 79.8 | 70.9 | 59.4 | 46.2 | 32.5 | 19.7 | 10.9 |
| | Stability (Li et al., 2019) | 1.636 | 96.5 | 94.6 | 91.6 | 87.2 | 80.7 | 71.7 | 60.5 | 47.0 | 33.4 | 20.6 | 11.2 |
| | SmoothAdv (Salman et al., 2019) | 1.779 | 95.8 | 93.9 | 90.6 | 86.5 | 80.8 | 73.7 | 64.6 | 53.9 | 43.3 | 32.8 | 22.2 |
| 1.00 | MACER (Zhai et al., 2020) | 1.598 | 91.6 | 88.1 | 83.5 | 77.7 | 71.1 | 63.7 | 55.7 | 46.8 | 38.4 | 29.2 | 20.0 |
| | Consistency (Jeong & Shin, 2020) | 1.738 | 95.0 | 93.0 | 89.7 | 85.4 | 79.7 | 72.7 | 63.6 | 53.0 | 41.7 | 30.8 | 20.3 |
| | SmoothMix (Jeong et al., 2021) | 1.820 | 93.7 | 91.6 | 88.1 | 83.5 | 77.9 | 70.9 | 62.7 | 53.8 | 44.8 | 36.6 | 28.9 |
| | **CAT-RS (Ours)** | **1.830** | 93.9 | 91.3 | 88.0 | 83.5 | 78.0 | **71.8** | **64.0** | **55.7** | **47.3** | **38.9** | **29.8** |

(a) $\sigma = 0.25$     (b) $\sigma = 0.50$     (c) $\sigma = 1.00$

Figure 3: Comparison of approximate certified accuracy for various training methods on MNIST. The sharp drop of certified accuracy in each plot is due to a strict upper bound in radius that CERTIFY can output for a given $\sigma$, $N = 100,000$, and $\alpha = 0.001$.

also perform a thorough ablation study on our loss design and confirm that (a) both of our proposed components have effectiveness to improve the certified robustness, and (b) the single hyperparameter $\lambda$ (10) between the two losses can balance the trade-off between robustness and accuracy.

**Training setups.** For a fair comparison, we follow the training setup considered in most of the previous works to compare the performance of the smoothed classifiers (Cohen et al., 2019; Zhai et al., 2020; Jeong & Shin, 2020; Jeong et al., 2021): specifically, we mainly consider LeNet (LeCun et al., 1998) and ResNet-110 (He et al., 2016) for MNIST and CIFAR-10, respectively, and consider three different scenarios of $\sigma = 0.25, 0.5$, and $1.0$ for randomized smoothing. We train each model for 150 epochs with stochastic gradient descent (SGD) with a momentum of 0.9. The learning rates are initialized to 0.01 for MNIST and 0.1 for CIFAR-10, and decreased by a factor of 0.1 in every 50 epochs. We apply the same $\sigma$ for both training and evaluation. More details on the training setups, *e.g.*, hyperparameters, can be found in Appendix B.

**Baselines.** We compare our method with an extensive list of baseline methods in the literature for training smoothed classifiers: (a) *Gaussian training* (Cohen et al., 2019) simply trains a classifier with Gaussian augmentation (5); (b) *Stability training* (Li et al., 2019) adds a cross-entropy term between the logits from clean and noisy images; *SmoothAdv* (Salman et al., 2019) employs adversarial training for smoothed classifiers (9); (d) *MACER* (Zhai et al., 2020) adds a regularization that aims to maximize a soft approximation of certified radius; (e) *Consistency* (Jeong & Shin, 2020) regularizes the variance of confidences over Gaussian noise; (f) *SmoothMix* (Jeong et al., 2021) proposes a mixup-based (Zhang et al., 2018) adversarial training for smoothed classifiers. Whenever possible, we use the pre-trained models publicly released by the authors to reproduce the results.

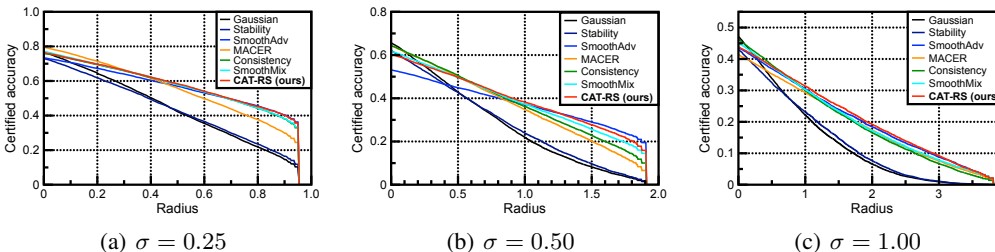

(a) $\sigma = 0.25$  (b) $\sigma = 0.50$  (c) $\sigma = 1.00$

Figure 4: Comparison of approximate certified accuracy for various training methods on CIFAR-10. The sharp drop of certified accuracy in each plot is due to a strict upper bound in radius that CERTIFY can output for a given $\sigma$, $N = 100,000$, and $\alpha = 0.001$.

**Evaluation metrics.** We follow the standard evaluation protocol for smoothed classifiers (Salman et al., 2019; Zhai et al., 2020; Jeong & Shin, 2020; Jeong et al., 2021): specifically, Cohen et al. (2019) have proposed a practical Monte Carlo based certification procedure, namely CERTIFY, that returns the prediction of $\hat{f}$ and a lower bound of certified radius, $\mathrm{CR}(f, \sigma, x)$, over the randomness of $n$ samples with probability at least $1 - \alpha$, or abstains the certification. Based on CERTIFY, we consider two major evaluation metrics: (a) the *average certified radius* (ACR) (Zhai et al., 2020): the average of certified radii on the test dataset $\mathcal{D}_{\text{test}}$ while assigning incorrect samples as 0, namely $\mathrm{ACR} := \frac{1}{|\mathcal{D}_{\text{test}}|} \sum_{(x,y) \in \mathcal{D}_{\text{test}}} [\mathrm{CR}(f, \sigma, x) \cdot \mathbb{1}_{\hat{f}(x)=y}]$, and (b) the *approximate certified test accuracy* at $r$: the fraction of the test dataset which CERTIFY classifies correctly with the radius larger than $r$ without abstaining. We use $n = 100,000$, $n_0 = 100$, and $\alpha = 0.001$ for CERTIFY, following the previous works (Cohen et al., 2019; Salman et al., 2019; Jeong & Shin, 2020; Jeong et al., 2021).

## 4.1 RESULTS ON MNIST

We compare the certified robustness of the smoothed classifiers on MNIST from our method to those from other baselines in Table 1. We also present in Figure 3 the plots of the approximate certified accuracy across varying $r$ for $\sigma \in \{0.25, 0.5, 1.0\}$. Overall, the results show that our method of CAT-RS clearly surpasses all the other baselines in terms of ACR: *i.e.*, our method could better balance between the clean accuracy and robustness. For $\sigma = 0.25$, we notice that some baselines, *i.e.*, SmoothAdv and SmoothMix, already achieve a reasonably saturated level of ACR: even in this trivial task, our method could further improve the robust accuracy at $r = 0.75$ as $96.8\% \rightarrow 97.0\%$. In more challenging cases of $\sigma = 0.5$ and $\sigma = 1.0$, on the other hand, the improvements from CAT-RS in ACR become more evident as $\sigma$ increases: *e.g.*, at $\sigma = 1.0$, compared to SmoothMix (the best-performing baseline), CAT-RS could improve the certified accuracy at $r = 2.50$ by $28.9\% \rightarrow 29.8\%$ while even improving the clean accuracy (*i.e.*, certified accuracy at $r = 0.0$) by $93.7\% \rightarrow 93.9\%$. This means that our proposed CAT-RS can be more effective at challenging tasks, where it is more likely that a given classifier gets a more diverse confidence distribution for the training samples, so that our proposed confidence-aware training can better play its role.

**Accuracy-robustness trade-off.** To further validate that our method can exhibit a better trade-off between accuracy and robustness compared to other methods, we additionally compare the performance trends between clean accuracy and certified accuracy at $r = 2.0$ as we vary a hyperparameter to control the trade-off, *e.g.*, $\lambda$ (10) in case of our method. We use $\sigma = 1.0$ for this experiment. We choose Consistency (Jeong & Shin, 2020) and SmoothMix (Jeong et al., 2021) for this comparison, considering that they also offer a single hyperparameter (namely $\lambda$ and $\eta$, respectively) for the balance between accuracy and robustness similar to our method, while both generally achieve good performances among the baselines considered. The results plotted in Figure 5 clearly show that CAT-RS indeed exhibits a higher trade-off frontier compared to both methods, which confirms the effectiveness of our method. More detailed results can be found in Appendix C.

## 4.2 RESULTS ON CIFAR-10

Table 2 shows the performance of the baselines and our model on CIFAR-10. In Figure 4, we also plot the approximate certified accuracy over the range of $r$ for $\sigma \in \{0.25, 0.5, 1.0\}$. In this experiment, for each $\sigma$, the baseline models are individually chosen based on its "best" ACR, *i.e.*,

Table 2: Comparison of ACR and approximate certified test accuracy (%) on CIFAR-10. For each column, we set our result bold-faced whenever the value improves the Gaussian baseline. We mark the highest and lowest values of certified accuracy at each radius in blue and red colors, respectively.

| σ | Methods | ACR | 0.00 | 0.25 | 0.50 | 0.75 | 1.00 | 1.25 | 1.50 | 1.75 | 2.00 | 2.25 | 2.50 |
|---|---|---|---|---|---|---|---|---|---|---|---|---|---|
| | Gaussian (Cohen et al., 2019) | 0.424 | 76.6 | 61.2 | 42.2 | 25.1 | 0.0 | 0.0 | 0.0 | 0.0 | 0.0 | 0.0 | 0.0 |
| | Stability (Li et al., 2019) | 0.420 | 73.0 | 58.9 | 42.9 | 26.8 | 0.0 | 0.0 | 0.0 | 0.0 | 0.0 | 0.0 | 0.0 |
| | SmoothAdv (Salman et al., 2019) | 0.544 | 73.4 | 65.6 | 57.0 | 47.5 | 0.0 | 0.0 | 0.0 | 0.0 | 0.0 | 0.0 | 0.0 |
| 0.25 | MACER (Zhai et al., 2020) | 0.531 | 79.5 | 69.0 | 55.8 | 40.6 | 0.0 | 0.0 | 0.0 | 0.0 | 0.0 | 0.0 | 0.0 |
| | Consistency (Jeong & Shin, 2020) | 0.552 | 75.8 | 67.6 | 58.1 | 46.7 | 0.0 | 0.0 | 0.0 | 0.0 | 0.0 | 0.0 | 0.0 |
| | SmoothMix (Jeong et al., 2021) | 0.553 | 77.1 | 67.9 | 57.9 | 46.7 | 0.0 | 0.0 | 0.0 | 0.0 | 0.0 | 0.0 | 0.0 |
| | **CAT-RS (Ours)** | **0.557** | 76.2 | **68.1** | **58.4** | **47.4** | 0.0 | 0.0 | 0.0 | 0.0 | 0.0 | 0.0 | 0.0 |
| | Gaussian (Cohen et al., 2019) | 0.525 | 65.7 | 54.9 | 42.8 | 32.5 | 22.0 | 14.1 | 8.3 | 3.9 | 0.0 | 0.0 | 0.0 |
| | Stability (Li et al., 2019) | 0.531 | 62.1 | 52.6 | 42.7 | 33.3 | 23.8 | 16.1 | 9.8 | 4.7 | 0.0 | 0.0 | 0.0 |
| | SmoothAdv (Salman et al., 2019) | 0.717 | 53.1 | 49.2 | 44.9 | 41.0 | 37.2 | 33.2 | 29.1 | 24.0 | 0.0 | 0.0 | 0.0 |
| 0.50 | MACER (Zhai et al., 2020) | 0.691 | 64.2 | 57.5 | 49.9 | 42.3 | 34.8 | 27.6 | 20.2 | 12.6 | 0.0 | 0.0 | 0.0 |
| | Consistency (Jeong & Shin, 2020) | 0.720 | 64.3 | 57.5 | 50.6 | 43.2 | 36.2 | 29.5 | 22.8 | 16.1 | 0.0 | 0.0 | 0.0 |
| | SmoothMix (Jeong et al., 2021) | 0.737 | 61.8 | 55.9 | 49.5 | 43.3 | 37.2 | 31.7 | 25.7 | 19.8 | 0.0 | 0.0 | 0.0 |
| | **CAT-RS (Ours)** | **0.752** | 60.2 | **55.0** | **49.7** | **43.8** | **38.2** | **33.2** | **27.7** | **22.0** | 0.0 | 0.0 | 0.0 |
| | Gaussian (Cohen et al., 2019) | 0.511 | 47.1 | 40.9 | 33.8 | 27.7 | 22.1 | 17.2 | 13.3 | 9.7 | 6.6 | 4.3 | 2.7 |
| | Stability (Li et al., 2019) | 0.514 | 43.0 | 37.8 | 32.5 | 27.5 | 23.1 | 18.8 | 14.7 | 11.0 | 7.7 | 5.2 | 3.1 |
| | SmoothAdv (Salman et al., 2019) | 0.790 | 43.7 | 40.3 | 36.9 | 33.8 | 30.5 | 27.0 | 24.0 | 21.4 | 18.4 | 15.9 | 13.4 |
| 1.00 | MACER (Zhai et al., 2020) | 0.744 | 41.4 | 38.5 | 35.2 | 32.3 | 29.3 | 26.4 | 23.4 | 20.2 | 17.4 | 14.5 | 12.1 |
| | Consistency (Jeong & Shin, 2020) | 0.756 | 46.3 | 42.2 | 38.1 | 34.3 | 30.0 | 26.3 | 22.9 | 19.7 | 16.6 | 13.8 | 11.3 |
| | SmoothMix (Jeong et al., 2021) | 0.773 | 45.1 | 41.5 | 37.5 | 33.8 | 30.2 | 26.7 | 23.4 | 20.2 | 17.2 | 14.7 | 12.1 |
| | **CAT-RS (Ours)** | **0.812** | 43.9 | **40.9** | **37.6** | **34.4** | **31.3** | **28.0** | **25.1** | **22.3** | **19.3** | **16.6** | **13.9** |

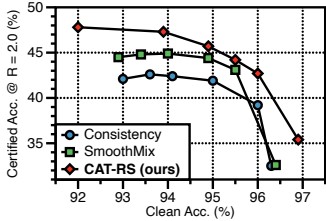

Figure 5: Comparison of clean *vs.* certified accuracy at $r = 2.0$ on MNIST ($\sigma = 1.0$) for methods those offer trade-off control.

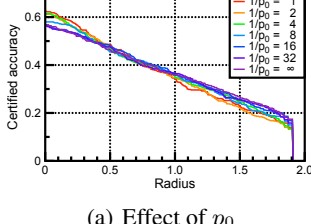

(a) Effect of $p_0$

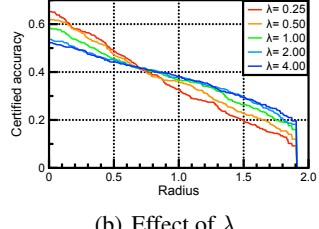

(b) Effect of $\lambda$

Figure 6: Comparison of certified accuracy of CAT-RS ablations on CIFAR-10. We use ResNet-20 for ablation study and plot the results at $\sigma = 0.5$. More results can be found in Appendix D.

the hyperparameter of the same baseline may vary over $\sigma$. For example, we choose the SmoothAdv baseline as the best model from the hundreds of hyperparameter configurations those examined by Salman et al. (2019) for each $\sigma$. Overall, our method of CAT-RS achieves a significant improvement of ACR compared to the baselines. In case of $\sigma = 0.25$ and $\sigma = 0.5$, for example, CAT-RS clearly offers a better trade-off between the clean accuracy and robustness compared to SmoothAdv, in a sense that (a) they achieve similar certified accuracy at large $r$, yet (b) CAT-RS could maintain much higher clean accuracy, *e.g.*, $53.2\% \rightarrow 60.2\%$ in case of $\sigma = 0.5$. For $\sigma = 1.0$, the ACR of our method significantly surpasses the previous best model, SmoothMix, by $0.773 \rightarrow 0.812$. As in MNIST, the improvement of CAT-RS is most evident in $\sigma = 1.0$, demonstrating the effectiveness of confidence-aware training. Furthermore, we observe that the performance gap between our method and the previous best model is more significant in CIFAR-10 because the higher complexity of CIFAR-10 compared to MNIST makes the confidence information more critical.

## 4.3 ABLATION STUDY

We also conduct an ablation study to further analyze individual effectiveness of the design components in our method. Unless otherwise specified, we use ResNet-20 (He et al., 2016) throughout this section and test it on a subsampled CIFAR-10 of size 500. We assume $\lambda = 1.0$ and $p_0 = \frac{1}{10}$ by default. We report the detailed results for this study in Appendix D.

**Effect of $p_0$.** We introduce a hyperparameter $p_0$ in (7) to control how much we constrain the confidence of "hard" noises, so that $p_0 \rightarrow 1$ would lead the proposed $L^{\text{low}}$ (7) into the standard

Table 3: Comparison of ACR and approximate certified test accuracy (%) varying components of CAT-RS on CIFAR-10. We assume $\sigma = 0.5$ in this experiment. The best ACR is bold-faced.

| Method (CIFAR-10) | ACR | 0.00 | 0.25 | 0.50 | 0.75 | 1.00 | 1.25 | 1.50 | 1.75 |
|---|---|---|---|---|---|---|---|---|---|
| Gaussian | 0.532 | 68.2 | 54.8 | 41.0 | 31.8 | 23.0 | 15.4 | 9.2 | 4.0 |
| (a) $L^{\texttt{low}}$ only | 0.552 | 67.0 | 55.4 | 44.8 | 33.0 | 24.6 | 16.8 | 9.4 | 5.0 |
| (b) $L^{\texttt{high}}$ only | 0.687 | 52.8 | 48.2 | 44.2 | 39.2 | 35.2 | 31.8 | 26.4 | 22.0 |
| (c) $L^{\texttt{low}} + \lambda \cdot L^{\texttt{high}}$ | 0.712 | 57.0 | 51.8 | 47.0 | 41.6 | 36.6 | 32.0 | 25.6 | 20.2 |
| (d) $L^{\texttt{high}} \rightarrow L^{\texttt{avg.max}}$ (11) | 0.692 | 60.2 | 54.0 | 46.8 | 41.0 | 35.8 | 28.6 | 22.8 | 17.4 |
| $L^{\texttt{CAT-RS}}$ (**Ours;** (10)) | **0.717** | 58.4 | 52.2 | 45.4 | 41.2 | 36.6 | 33.6 | 26.4 | 20.2 |

**Gaussian training.** To verify its effectiveness, we conduct an experiment comparing the certified robustness of models with different $p_0 \in \{0, \frac{1}{32}, \frac{1}{16}, \frac{1}{8}, \frac{1}{4}, \frac{1}{2}, 1\}$ on $\sigma \in \{0.25, 0.5, 1.0\}$: Table 5 in Appendix D summarizes the results. Overall, we observe that setting $p_0 < 1$ is clearly beneficial to improve ACR for all the $\sigma$ considered, *e.g.*, in case of $\sigma = 1.0$ the method achieve the best ACR when $p_0 = \frac{1}{16}, \frac{1}{32}$, and this confirms that our proposed loss of $L^{\texttt{low}}$ (7) alone is superior to the Gaussian training even without the effect from $L^{\texttt{high}}$ (8). From the Figure 6(a) which plots the results at $\sigma = 0.5$, one can observe that letting $p_0 \rightarrow 0$ leads the classifier to achieve a better robust accuracy with a relatively little degradation in the clean accuracy.

**Effect of $\lambda$.** By the definition in (10), $\lambda$ controls the contribution of $L^{\texttt{high}}$. We evaluate the impact of $\lambda$ here, and the results are shown in Figure 6(b). We compare the performance of the models, varying $\lambda \in \{0.25, 0.5, 1.0, 2.0, 4.0\}$ on $\sigma = 0.5$. As expected, we observe that $\lambda$ balances the trade-off between robustness and clean accuracy; as $\lambda$ increases, robustness increases while clean accuracy decreases. Also, in Table 7 in Appendix C, we verify that CAT-RS offers more effective trade-off between robustness and the clean accuracy than other methods. Further details can be found in Table 6 in Appendix D.

**Loss design.** Our loss design in (10) combines several important ideas proposed in Section 3, and here we validate that each of the components has an individual effect in improving the certified robustness. In Table 3, we compare several variants of our proposed CAT-RS loss (10), namely (a) using only $L^{\texttt{low}}$ (7), (b) using only $L^{\texttt{high}}$ (8), and (c) $L^{\texttt{low}} + \lambda \cdot L^{\texttt{high}}$, *i.e.*, not applying the masking condition $\mathbb{1}[\hat{p}_f(x, y) = 1]$ to $L^{\texttt{high}}$. We also consider (d) a variant of $L^{\texttt{high}}$, where the outer max operation over noise samples (8) is replaced by the average, *i.e.*, we define:

$$L^{\texttt{avg.max}} := \frac{1}{M} \sum_i \left( \max_{\|\delta_i^* - \delta_i\|_2 \leq \varepsilon} \mathbb{CE}(F(x + \delta_i^*), y) \right). \tag{11}$$

We report the experimental results on $\sigma = 0.5$. Overall, CAT-RS achieves the best ACR and approximate certified accuracy among the variants. We observe that $L^{\texttt{low}}$ alone improves Gaussian baseline, which highlights the importance of the confidence information for robust training. Also, $L^{\texttt{high}}$ alone increases the robustness dramatically, while it sacrifices the clean accuracy. Combining $L^{\texttt{low}}$ and $L^{\texttt{high}}$ achieves better results by balancing robustness and accuracy. CAT-RS further improves the overall performance by simply masking out $L^{\texttt{high}}$ for low-confidential predictions: *i.e.*, when the confidence is not high enough, learning challenging adversarial samples harm the performance. By comparing the results made by using $L^{\texttt{avg.max}}$ and $L^{\texttt{high}}$, we verify that it is more helpful for a model to utilize the most challenging adversarial examples instead of averaging them.

## 5 CONCLUSION

This paper explores a close relationship between confidence and robustness, a natural property of smoothed classifiers yet neural networks cannot currently offer. We have successfully leveraged this relationship to relax the hard-to-compute metric of adversarial robustness into an easier concept of prediction confidence. Consequently, we could propose a practical robust training method that enables a sample-level control of adversarial robustness, which has been difficult without heuristics in a conventional belief. We believe our work could be a useful step for the future research on exploring the interesting connection between adversarial robustness and *confidence calibration* (Guo et al., 2017; Lee et al., 2018) through the framework of randomized smoothing.

ETHICS STATEMENT

Deploying deep learning based systems into the real-world, especially when they are of security-concerned (Caruana et al., 2015; Yurtsever et al., 2020), still poses many potential risks for both companies and customers, and we researchers are responsible to make this technology more reliable through research towards *AI safety* (Amodei et al., 2016). *Adversarial robustness* is one of the central parts of this direction, and we believe our research on certified robustness can be a useful step towards building a practical yet secure deep learning system. Nevertheless, one should also recognize that adversarial robustness is still a bare minimum requirement for reliable deep learning, and the future research should address how to extend this restrictive notion of robustness into other challenging setups, *e.g.*, corruption robustness (Hendrycks et al., 2020) and unrestricted attacks (Bhattad et al., 2020), just to name a few, to establish a practical sense of security for practitioners.

REPRODUCIBILITY STATEMENT

For the best practices to maintain the reproducibility of this paper, we have conducted our experiments on publicly available datasets, with the detailed descriptions for pre-processing in Appendix B.1. We have documented a thorough specification on the experimental details, *e.g.*, training setups, baselines, and evaluation metrics, in Section 4. We have also fully specified all the hyperparameters considered to reproduce the baselines as well as our results in Appendix B.2. The code to reproduce our results will be publicly available, as well as the pre-trained models for our method.

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

## A    Training procedure of CAT-RS

---

**Algorithm 1** Confidence-Aware Training for Randomized Smoothing (CAT-RS)

---

**Require:** training sample $(x, y)$. smoothing factor $\sigma$. number of noise samples $M > 0$. truncation probability $p_0 \in (0, 1]$, regularization strength $\lambda > 0$. attack norm $\varepsilon > 0$.

---

1: Sample $\delta_1, \cdots, \delta_M \sim \mathcal{N}(0, \sigma^2 I)$
2: $\hat{p}_f \leftarrow \frac{1}{M} \sum_i \mathbb{1}[f(x + \delta_i) = y]$
3: // Compute the bottom-$K$ loss
4: Sample $K \sim \text{Bin}(M, \hat{p}_f)$
5: $K^+ \leftarrow \max(1, K)$
6: **for** $i = 1$ **to** $M$ **do**
7:     $L_i \leftarrow \mathbb{CE}(F(x + \delta_i), y)$
8: **end for**
9: $L^\pi_{1:M} \leftarrow \texttt{argsort}(L_{1:M})$
10: $L^{\texttt{low}} \leftarrow \frac{1}{M}(\sum_{i=1}^{K^+} L^\pi_i + \sum_{i=K^++1}^{M} (L^\pi_i + \log p_0)^+)$
11: // Compute the worst-case loss
12: **for** $i = 1$ **to** $M$ **do**
13:     $\delta^*_i \leftarrow \arg\max_{\|\delta^*_i - \delta_i\| \leq \varepsilon} \mathbb{CE}(F(x + \delta^*_i), y)$
14: **end for**
15: $L^{\texttt{high}} \leftarrow \max_i \mathbb{CE}(F(x + \delta^*_i), y)$
16: // Compute the CAT-RS loss
17: $L^{\texttt{CAT-RS}} \leftarrow L^{\texttt{low}} + \lambda \cdot \mathbb{1}[\hat{p}_f = 1] \cdot L^{\texttt{high}}$

---

## B  EXPERIMENTAL DETAILS

### B.1  DATASETS

**MNIST** (LeCun et al., 1998) consists of 70,000 gray-scale hand-written digit images of size 28×28, 60,000 for training and 10,000 for testing, where each is labeled to one value between 0 and 9. We do not perform any pre-processing except for normalizing the range of each pixel from 0-255 to 0-1. The dataset can be downloaded at `http://yann.lecun.com/exdb/mnist/`.

**CIFAR-10** (Krizhevsky, 2009) consists of 60,000 RGB images of size 32×32 pixels, 50,000 for training and 10,000 for testing, where each is labeled to one of 10 classes. We use the standard data-augmentation scheme of random horizontal flip and random translation up to 4 pixels, following the practice of other baselines (Cohen et al., 2019; Salman et al., 2019; Zhai et al., 2020; Jeong & Shin, 2020; Jeong et al., 2021). We also normalize the images in pixel-wise by the mean and the standard deviation calculated from the training set. The full dataset can be downloaded at `https://www.cs.toronto.edu/~kriz/cifar.html`.

### B.2  HYPERPARAMETERS

**Stability training** (Li et al., 2019) uses a single hyperparameter $\gamma$ to control the relative strength of the regularization term. We fix $\gamma = 2$ for MNIST experiments. For CIFAR-10 experiments, $\gamma = 2$ is used for $\sigma = 0.25, 0.5$, and $\gamma = 1$ is used for $\sigma = 1.0$.

**SmoothAdv** (Salman et al., 2019) uses three major hyperparameters to perform the projected gradient descent: namely, the attack radius in terms of $\ell_2$-norm $\varepsilon$, the number of PGD steps $T$, and the number of noises $m$. In our experiments, we fix $T = 10$. For MNIST experiments, we fix $\varepsilon = 1.0$ and $m = 4$ as well. In case of CIFAR-10, on the other hand, we report the results chosen among the list of "best" configurations for each noise level which are previously searched by Salman et al. (2019): specifically, we report the results of $\varepsilon = 1.0$ and $m = 4$ for $\sigma = 0.25$, and $\varepsilon = 2.0$ and $m = 2$ for $\sigma = 0.5, 1.0$. When SmoothAdv is used, we adopt the *warm-up* strategy, *i.e.*, we initially set $\varepsilon = 0.0$ and linearly increase to the target value of $\varepsilon$ for 10-epochs.

**MACER** (Zhai et al., 2020) introduces four hyperparameters: namely, the number of noises $k$, the coefficient for the regularization term $\lambda$, the clamping parameter for maximizing the certified radius $\gamma$, and the temperature scaling parameter $\beta$. For the MNIST experiments, we use $k = 16, \gamma = 8.0, \beta = 16.0$, and $\lambda = 16.0$ when $\sigma = 0.25, 0.5$, following the configurations in Zhai et al. (2020). When $\sigma = 1.0$, for the training to succeed, we set $\lambda = 6.0$. For the CIFAR-10 experiments, we follow the original configurations used by Zhai et al. (2020). We set $k = 16, \gamma = 8.0$, and $\beta = 16.0$. $\lambda$ is set to be 12.0 and 4.0 for $\sigma = 0.25$ and 0.5, respectively. For $\sigma = 1.0$, the training starts with $\lambda = 0$ until the first learning rate decay and we set $\lambda = 12.0$ thereafter.

**Consistency** (Jeong & Shin, 2020) uses two hyperparameters: namely, the coefficient for the consistency term $\eta$ and those for the entropy term $\gamma$. We report the best results in terms of ACR among those reported by Jeong & Shin (2020) varying $\eta$. Following the original practice, we fix $\gamma = 0.5$ throughout our experiments. For MNIST, we use $\lambda = 10$ for $\sigma = 0.25$ and $\lambda = 5$ for other noises. For the CIFAR-10 experiments, we use $\lambda = 20$ for $\sigma = 0.25$ and $\lambda = 10$ for other noises.

**SmoothMix** (Jeong et al., 2021) introduces four hyperparameters: namely, the coefficient for the mixup loss $\eta$, the step size for adversarial attack $\alpha$, the number of steps for adversarial attack $T$, and the number of noises $T$. For the MNIST experiments, we fix $\eta = 5.0, \alpha = 1.0$, and $m = 4$. $T = 2, 4, 8$ are used for the models with $\sigma = 0.25, 0.5, 1.0$, respectively. For the CIFAR-10 experiments, we again report the best result among those reported from Jeong et al. (2021): *i.e.*, we fix $\eta = 5.0, m = 2$, and $T = 4$, and use $\alpha = 0.5, 1.0, 2.0$ for $\sigma = 0.25, 0.5, 1.0$, respectively. The "one-step adversary" is used for $\sigma = 0.5, 1.0$ to follow the "best" configurations.

**CAT-RS (Ours).** CAT-RS uses two main hyperparameters: namely, the coefficient for "worst-case" loss $\lambda$ and clamping parameter for maximizing the confidence of "hard" samples $p_0$. For the MNIST experiments, we use the fixed configuration of $\lambda = 1.0$ and $p_0 = \frac{1}{5}$. For the CIFAR-10 experiments, we use $\lambda = 0.5, p_0 = \frac{1}{20}$ for $\sigma = 0.25$, and $\lambda = 2.0, p_0 = \frac{1}{10}$ for $\sigma = 0.5$ and 1.0.

# C   COMPARISON OF ACCURACY-ROBUSTNESS TRADE-OFF

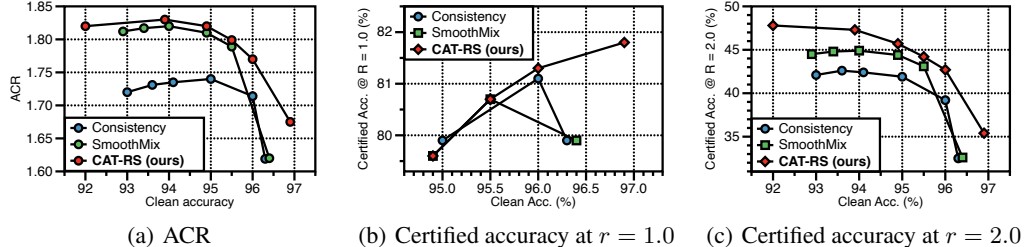

(a) ACR          (b) Certified accuracy at $r = 1.0$     (c) Certified accuracy at $r = 2.0$

Figure 7: (a): Comparison of the trends between the clean accuracy *vs.* (a) ACR, (b) the certified accuracy at $r = 1.0$, and (c): the certified accuracy at $r = 2.0$, that each method exhibits as varying its hyperparameter. We assume MNIST dataset with $\sigma = 1.0$ for this experiment.

Table 4: Comparison of ACR and approximate certified test accuracy on MNIST for varying hyperparameters of three different methods: Consistency, SmoothMix, and CAT-RS (ours). We assume $\sigma = 1.0$ in this experiment. "Gaussian" indicates the baseline Gaussian training. Consistency and SmoothMix degenerates to Gaussian when their hyperparameter is set to 0.

| Methods | Setups | ACR | 0.00 | 0.50 | 1.00 | 1.50 | 2.00 | 2.50 |
|---|---|---|---|---|---|---|---|---|
| Gaussian | - | 1.620 | 96.4 | 91.4 | 79.9 | 59.6 | 32.6 | 10.8 |
| Consistency | $\lambda = 1$ | 1.714 | 96.0 | 91.2 | 81.1 | 63.5 | 39.2 | 16.2 |
| | $\lambda = 5$ | 1.740 | 95.0 | 89.7 | 79.9 | 63.7 | 41.9 | 20.0 |
| | $\lambda = 10$ | 1.735 | 94.1 | 88.6 | 78.5 | 62.8 | 42.4 | 22.1 |
| | $\lambda = 15$ | 1.731 | 93.6 | 87.7 | 77.8 | 62.3 | 42.6 | 22.9 |
| | $\lambda = 20$ | 1.720 | 93.0 | 86.6 | 77.1 | 61.6 | 42.1 | 23.4 |
| | $\lambda = 25$ | 1.226 | 73.2 | 64.4 | 53.9 | 42.4 | 27.4 | 14.5 |
| SmoothMix | $\eta = 1$ | 1.789 | 95.5 | 90.5 | 80.7 | 64.1 | 43.1 | 24.1 |
| | $\eta = 2$ | 1.810 | 94.9 | 89.7 | 79.6 | 63.8 | 44.4 | 26.6 |
| | $\eta = 4$ | 1.820 | 94.0 | 88.4 | 78.3 | 63.0 | 44.9 | 28.7 |
| | $\eta = 8$ | 1.817 | 93.4 | 87.5 | 77.3 | 62.4 | 44.8 | 29.3 |
| | $\eta = 16$ | 1.812 | 92.9 | 86.7 | 76.6 | 61.8 | 44.5 | 29.6 |
| **CAT-RS (Ours)** | $\lambda = 0.00$ | 1.675 | 96.9 | 92.1 | 81.8 | 62.7 | 35.4 | 12.4 |
| | $\lambda = 0.12$ | 1.770 | 96.0 | 91.3 | 81.3 | 64.7 | 42.7 | 20.9 |
| | $\lambda = 0.25$ | 1.799 | 95.5 | 90.5 | 80.7 | 64.8 | 44.2 | 24.5 |
| | $\lambda = 0.50$ | 1.820 | 94.9 | 89.5 | 79.6 | 64.5 | 45.7 | 27.4 |
| | $\lambda = 1.00$ | 1.830 | 93.9 | 88.0 | 78.0 | 64.0 | 47.3 | 29.8 |
| | $\lambda = 2.00$ | 1.820 | 92.0 | 85.4 | 75.5 | 62.7 | 47.8 | 31.8 |
| | $\lambda = 4.00$ | 1.788 | 89.0 | 82.0 | 72.8 | 62.1 | 48.7 | 32.6 |

# D  DETAILED RESULTS ON ABLATION STUDY

Table 5: Comparison of ACR and approximate certified test accuracy (%) varying $p_0$ on CIFAR-10. For each $\sigma$, we set ACR bold-faced when it achieves the best among variants.

| $\sigma$ | Setups | ACR | Certified accuracy (%) | | | | | | | | | |
|---|---|---|---|---|---|---|---|---|---|---|---|---|
| | | | 0.0 | 0.25 | 0.5 | 0.75 | 1.0 | 1.25 | 1.5 | 1.75 | 2.0 | 2.25 |
| 0.25 | $1/p_0 = 1.0$ | 0.536 | 71.6 | 64.2 | 56.8 | 47.8 | 0.0 | 0.0 | 0.0 | 0.0 | 0.0 | 0.0 |
| | $1/p_0 = 2.0$ | 0.524 | 69.4 | 62.4 | 56.0 | 46.2 | 0.0 | 0.0 | 0.0 | 0.0 | 0.0 | 0.0 |
| | $1/p_0 = 4.0$ | **0.538** | 71.8 | 63.6 | 56.0 | 48.0 | 0.0 | 0.0 | 0.0 | 0.0 | 0.0 | 0.0 |
| | $1/p_0 = 8.0$ | 0.522 | 67.2 | 61.8 | 54.8 | 47.4 | 0.0 | 0.0 | 0.0 | 0.0 | 0.0 | 0.0 |
| | $1/p_0 = 16.0$ | 0.515 | 67.6 | 61.2 | 52.8 | 46.6 | 0.0 | 0.0 | 0.0 | 0.0 | 0.0 | 0.0 |
| | $1/p_0 = 32.0$ | 0.523 | 68.4 | 62.2 | 55.2 | 45.8 | 0.0 | 0.0 | 0.0 | 0.0 | 0.0 | 0.0 |
| | $1/p_0 = \infty$ | 0.508 | 65.4 | 59.8 | 53.4 | 45.4 | 0.0 | 0.0 | 0.0 | 0.0 | 0.0 | 0.0 |
| 0.50 | $1/p_0 = 1.0$ | 0.700 | 62.4 | 56.6 | 48.4 | 40.2 | 34.4 | 28.8 | 23.0 | 18.2 | 0.0 | 0.0 |
| | $1/p_0 = 2.0$ | 0.690 | 61.4 | 54.4 | 47.0 | 40.0 | 34.6 | 28.2 | 22.2 | 16.8 | 0.0 | 0.0 |
| | $1/p_0 = 4.0$ | 0.707 | 61.8 | 54.6 | 46.8 | 40.8 | 35.0 | 30.0 | 24.6 | 18.6 | 0.0 | 0.0 |
| | $1/p_0 = 8.0$ | 0.705 | 58.0 | 54.6 | 45.6 | 41.8 | 36.4 | 31.0 | 24.0 | 18.8 | 0.0 | 0.0 |
| | $1/p_0 = 16.0$ | 0.713 | 56.4 | 53.2 | 48.0 | 41.4 | 36.2 | 30.6 | 25.6 | 20.8 | 0.0 | 0.0 |
| | $1/p_0 = 32.0$ | 0.707 | 56.4 | 52.6 | 47.6 | 39.6 | 36.4 | 30.4 | 25.4 | 21.2 | 0.0 | 0.0 |
| | $1/p_0 = \infty$ | **0.714** | 57.2 | 52.0 | 46.4 | 41.0 | 35.8 | 31.8 | 26.8 | 22.0 | 0.0 | 0.0 |
| 1.00 | $1/p_0 = 1.0$ | 0.716 | 48.2 | 41.2 | 37.6 | 32.0 | 27.4 | 24.0 | 20.6 | 17.8 | 15.0 | 13.0 |
| | $1/p_0 = 2.0$ | 0.732 | 47.0 | 44.0 | 38.2 | 33.2 | 26.4 | 22.6 | 20.6 | 17.4 | 16.2 | 13.4 |
| | $1/p_0 = 4.0$ | 0.743 | 46.4 | 41.4 | 37.0 | 32.8 | 27.8 | 25.4 | 21.2 | 18.6 | 16.0 | 13.4 |
| | $1/p_0 = 8.0$ | 0.787 | 45.8 | 42.0 | 38.6 | 33.8 | 30.4 | 25.6 | 22.6 | 20.0 | 17.8 | 15.4 |
| | $1/p_0 = 16.0$ | **0.815** | 45.2 | 41.2 | 39.4 | 34.2 | 30.8 | 28.2 | 23.8 | 19.6 | 17.4 | 15.6 |
| | $1/p_0 = 32.0$ | **0.815** | 45.2 | 41.6 | 38.0 | 33.8 | 31.2 | 28.4 | 24.2 | 20.4 | 17.0 | 15.8 |
| | $1/p_0 = \infty$ | 0.784 | 44.6 | 40.8 | 37.2 | 34.0 | 30.2 | 25.8 | 22.4 | 19.6 | 17.0 | 15.0 |

Table 6: Comparison of ACR and approximate certified test accuracy (%) for varying $\lambda$ on CIFAR-10. We assume $\sigma = 0.5$ in this experiment. For each column, we set the best value bold-faced.

| Setups | ACR | Certified accuracy (%) | | | | | | | |
|---|---|---|---|---|---|---|---|---|---|
| | | 0.0 | 0.25 | 0.5 | 0.75 | 1.0 | 1.25 | 1.5 | 1.75 |
| $\lambda = 0.25$ | 0.681 | **65.4** | **57.0** | **49.0** | 40.4 | 32.6 | 27.4 | 19.8 | 14.4 |
| $\lambda = 0.50$ | 0.704 | 62.2 | 56.4 | 47.2 | 39.6 | 36.0 | 29.6 | 23.0 | 17.0 |
| $\lambda = 1.00$ | 0.717 | 58.4 | 52.2 | 45.4 | **41.2** | 36.6 | 33.6 | 26.4 | 20.2 |
| $\lambda = 2.00$ | 0.719 | 53.6 | 49.0 | 45.0 | **41.2** | **37.8** | **34.2** | **29.6** | 22.8 |
| $\lambda = 4.00$ | **0.720** | 52.4 | 49.2 | 44.2 | 41.0 | 38.0 | 34.4 | 29.4 | **24.8** |

Table 7: Comparison of ACR and approximative certified test accuracy (%) for varying $M$ on CIFAR-10. We assume $\sigma = 0.5$ in this experiment.

| Setups | ACR | Certified accuracy (%) | | | | | | | |
|---|---|---|---|---|---|---|---|---|---|
| | | 0.0 | 0.25 | 0.5 | 0.75 | 1.0 | 1.25 | 1.5 | 1.75 |
| $M = 1$ | 0.667 | 63.6 | 55.4 | 47.0 | 38.6 | 33.4 | 25.8 | 19.0 | 14.8 |
| $M = 2$ | 0.689 | 60.8 | 54.4 | 46.8 | 40.8 | 35.0 | 28.8 | 22.8 | 17.6 |
| $M = 4$ | 0.717 | 58.4 | 52.2 | 45.4 | 41.2 | 36.6 | 33.6 | 26.4 | 20.2 |
| $M = 8$ | 0.708 | 55.4 | 49.8 | 45.0 | 40.8 | 36.4 | 33.0 | 27.4 | 21.8 |

