# OpenReview forum: "Confidence-aware Training of Smoothed Classifiers for Certified Robustness"
_ICLR.cc/2022/Conference — ICLR 2022 Submitted_

### Official Review · Reviewer_6h9p · 2021-10-18

**Correctness:** 4
**Technical Novelty And Significance:** 3
**Empirical Novelty And Significance:** 2
**Recommendation:** 3
**Confidence:** 4

**Main Review:**

This paper focuses on improving the certified robustness via training a better base classifier for randomized smoothing. The empirical analysis of "hard" and "easy" training samples is novel and innovative, which backs up the proposed losses. However, the experimental results do not provide strong empirical evidence on the effectiveness of the proposed method, and some important evaluation data, like results on ImageNet, running time statistics, etc, are not reported.

Strengths:
- The analysis, including the separation of "hard" and "easy" training samples and different treatments for them, is novel and interesting.
- The paper is very well written. The narration is coherent. The proposed method is clearly backed up by explanations.

Weaknesses:
- Based on the existing experimental results, the effectiveness of the proposed method is not apparent.

I appreciate that major baselines in this field are included in the experimental evaluation.

In terms of ACR, the proposed approach achieves state-of-the-art. However, the margin is not large enough, only except CIFAR-10 sigma=1.00 case. The authors claim that the method may be more significant on dataset with high complexity. Then IMHO, ImageNet results are inevitable to support this.

In terms of certified test accuracy, the proposed approach cannot achieve good benign accuracy or high certified accuracy under small radius though this is one of its primary goals (a better trade-off between robustness and accuracy). On CIFAR-10, the empirical performance seems to be better. However, the proposed method still cannot improve the accuracy under all radii compared with Gaussian (Cohen et al), and under most radii, the performance improvements are within 1.5%.

- Lack of ImageNet results.

All the compared baselines are evaluated on ImageNet. The high certified robustness on the standard ImageNet dataset is one of the unique strengths of randomized smoothing, and evaluation on ImageNet provides a more complete landscape on the method's performance on complex and large-scale datasets. Without ImageNet, it is hard to fairly evaluate the approach.

- Lack of training time statistics.

The M=4 intuitively would make the training process take > 4 times longer than Gaussian (Cohen et al), > 2 times longer than Consistency (Jeong et al), and about the same time as SmoothAdv (with m=4). Is it true? Then we may need to consider whether it is valuable to take a much longer time to obtain a small improvement.

Questions:
- On what set are the models evaluated?
Is the evaluation conducted on a uniformly picked subset of the full test set as other baselines? If so, is the set size 1000?
- Can you report the upper contour results across different sigma's for a clearer comparison?
- If the experiment results on ImageNet are provided, based on those results, I will re-evaluate this work.

Suggestion on the method:
- Maybe you can also try to assign different weights to the training samples. For example, if $p_f(x,y)$ is already high, we may sample more (e.g., set M=8/16 for each sample) to have a more precise estimation of $p_f(x,y)$, then assign higher weights to the sample with larger $p_f(x,y)$, since improvement on that sample would contribute more to ACR.
- In $L^{\mathtt{high}}$ loss, you first find $\delta^*$ then normalize them. Maybe you can incorporate the normalization inside - applying PGD on $\mathrm{normalized}(\delta^*)$.

**Summary Of The Paper:**

This paper proposes CAT-RS which combines two novel losses for training base classifiers for randomized smoothing. The two novel losses aim at preserving clean accuracy for hard samples and improving the certified radius for easy samples. In term of ACR (average certified radius), the method achieves state-of-the-art on MNIST and CIFAR-10.

**Summary Of The Review:**

I appreciate the analysis and the novelty of the proposed training method. However, the current loss design seems to be not effective enough, and there are a few important details missing in the experimental evaluation. Therefore, I do not lean towards acceptance. The authors may try to follow the proposed principle to improve the detail design of the current training approach.

---

> ### Author Response · Authors · 2021-11-23
> **Response to Reviewer 6h9p**
>
> We sincerely appreciate your thoughtful comments, efforts, and time. We respond to each of your questions and concerns one-by-one in what follows. All the responses will be carefully incorporated in the final draft.
>
> ---
> **Q1. Marginal improvements?**
>
> As you mentioned, our method consistently achieves state-of-the-arts across different setups, especially on the most challenging setup of $\sigma=1.0$ on CIFAR-10 (Table 2), but the gains may look less significant in easier setups like MNIST (Table 1) as their performances are likely to be saturated with recent state-of-the-art methods. Nevertheless, we emphasize two aspects that clearly show the superiority of CAT-RS even in the case of MNIST: (a) Figure 5 (in MNIST) still demonstrates that CAT-RS exhibits a significantly better operating points compared to prior arts under varying hyperparameters; (b) The table below further compares the variance of the ACRs in Table 1 over 5 runs: here, the results confirm that CAT-RS outperforms the other baselines significantly given the small variances. We will include the respective discussion in the final draft.
>
>
> | ACR (MNIST) | $\sigma=0.25$ | $\sigma=0.5$ | $\sigma=1.0$ |
> |:--------|:------------:|:------------:|:------------:|
> | Gaussian  | 0.9109 $\pm$ 0.0003 | 1.5581 $\pm$ 0.0016 | 1.6184 $\pm$ 0.0021 |
> | Consistency  | 0.9279 $\pm$ 0.0003 | 1.6549 $\pm$ 0.0011 | 1.7376 $\pm$ 0.0017 |
> | SmoothMix   | 0.9317 $\pm$ 0.0002 | 1.6932 $\pm$ 0.0007 | 1.8185 $\pm$ 0.0016 |
> | CAT-RS (ours)   | 0.9321 $\pm$ 0.0002 | 1.7001 $\pm$ 0.0006 | 1.8294 $\pm$ 0.0014 |
>
>
> ---
> **Q2. Lack of ImageNet results**
>
> Following your suggestion, we are currently working on ImageNet experiments, and we will include the results in the final draft.
>
> ---
> **Q3. Training time?**
>
> As shown in the below table on CIFAR-10 with a single TITAN X (Pascal) GPU, our method of CAT-RS takes as much time as SmoothAdv with $m=4$, while achieving a better ACR. We include the respective discussion in the final draft.
>
>
> | Methods  | Gaussian | Consistency (m=2) | SmoothAdv (m=4) | CAT-RS (ours) |
> |:--------|:------------:|:------------:|:------------:|:------------:|
> | Training time (hrs)  | 4.6 | 8.7 | 23.1 | 25.3 |
> | ACR ($\sigma=0.5$) | 0.525 | 0.720 | 0.717 | 0.752 |
>
>
> ---
> **Q4. Is the evaluation conducted on a subset, or the full test set?**
>
> We use the full test set of MNIST and CIFAR-10 for all the values in Table 1 and Table 2, following the practice used in previous works. We will clarify this point in the final draft.
>
> ---
> **Q5. Suggestions on the method details**
>
> Many thanks for the careful reading, and making constructive suggestions on the method details. We will consider all your suggestions and include the results in the final draft.

---

> > ### Comment · Reviewer_6h9p · 2021-11-24
> > **Thanks for authors' response**
> >
> > Thanks for the response. Most of my concerns are addressed. I appreciate the full test set evaluation on MNIST and CIFAR-10.
> >
> > However, I am still concerned about the practicality of the method given the long running time and not significant enough improvements (thanks for providing these additional statistics). Furthermore, the lack of ImageNet results is non-standard and there is no way to guarantee the ImageNet results will be added or will support the main claim of the paper.
> >
> > Therefore, the paper may not suitable for this venue at the current stage. I will keep my score unchanged and do hope the author could conduct a more comprehensive experimental evaluation for the next version.
> >
> > Best regards,
> > Reviewer 6h9p

---

### Official Review · Reviewer_dpjw · 2021-10-20

**Correctness:** 3
**Technical Novelty And Significance:** 1
**Empirical Novelty And Significance:** 1
**Recommendation:** 3
**Confidence:** 5

**Main Review:**

Strengths:
+The studied problem is important
+Paper is easy to follow

Weaknesses
-Novelty is limited
-Insufficient evaluation
-Missing important references

How do you make predictions during the training process, e.g., how do you obtain $\hat{p}_f $ in line 2 in Algorithm 1? Going forward, what’s the computational complexity of the proposed method?

The loss $L^{(high)}$ is only evaluated when  $\mathbb{1}[(\hat{p}_f=1)]$?  How many training samples with the associated generated noises make $(\hat{p}_f=1$. I thinks this is a very strong constrain on my end.

Why $p_0$ is limited to the range (0,1], instead of $(1, \infty)$?

The proposed method obtains comparable performance with the existing methods. How about the running time?

There is no result on the ImageNet.

The following paper also considers model’s confidence. Please discuss with it.

Kumar et al., “Certifying Confidence via Randomized Smoothing”


The following papers also derive certified robustness based on randomized smoothing

Wang et al. “Certified robustness of graph neural networks against adversarial structural perturbation via Randomized Smoothing”

Jia et al., “Certified Robustness for Top-k Predictions against Adversarial Perturbations via Randomized Smoothing”

Zhang et al., “Black-Box Certification with Randomized Smoothing: A Functional Optimization Based Framework”

Mohapatra et al., “Higher-Order Certification for Randomized Smoothing”

Kumar et al., “Certifying Confidence via Randomized Smoothing”

Kumar et al., “Curse of Dimensionality on Randomized Smoothing for Certifiable Robustness ”

Fischer  et al., “Certified Defense to Image Transformations via Randomized Smoothing”

Lee et al., “Tight Certificates of Adversarial Robustness for Randomly Smoothed Classifiers”


**Summary Of The Paper:**

The paper studies certified robustness via randomized smoothing (RS). RS has a fundamental accuracy and robustness tradeoff. The authors aim to enhance such tradeoff through a sample-wise control of robustness over the training samples. In particular, the authors investigate the correspondence between robustness and prediction confidence of smoothed classifiers and design a new loss function. The proposed method is evaluated on MNIST and CIFAR10.

**Summary Of The Review:**

The studied problem is important and paper is easy to follow. However, the novelty is limited and the evaluation is insufficient.

---

> ### Author Response · Authors · 2021-11-23
> **Response to Reviewer dpjw**
>
> We sincerely appreciate your thoughtful comments, efforts, and time. We respond to each of your questions and concerns one-by-one in what follows. All the responses will be carefully incorporated in the final draft.
>
> ---
> **Q1. How to compute $\hat{p}_f$ (Line 2 in Algorithm 1)? How about its computational complexity, and the running time?**
>
> As stated at Line 2 in Algorithm 1, $\hat{p}_f(x,y)$ (for a given $x$ and $y$) is computed by averaging the indicators whether $f(x + \delta_i) = y$ over $M$ i.i.d. Gaussian noises $\delta_1, \delta_2, …, \delta_M$ (the noises are re-used to compute low/high loss in Equation (7) and (8) as well), i.e., the fraction of $\delta_i$’s of $f(x+\delta_i)=y$: so it requires $O(M)$ times of forward passes per sample. Nevertheless, we note that using multiple noises (and multiple forwards accordingly) is a common design for state-of-the-art training methods of smoothed classifiers, e.g., all the baselines considered except Gaussian and Stability also require O(M) forwards. The table below compares the training statistics on CIFAR-10 with a single TITAN X (Pascal) GPU, and it shows that our method of CAT-RS takes as much time as SmoothAdv with $m=4$, while achieving a significantly better ACR. We add the respective discussion in the final draft.
>
> | Methods  | Gaussian | Consistency ($m=2$) | SmoothAdv ($m=4$) | CAT-RS (ours) |
> |:--------|:------------:|:------------:|:------------:|:------------:|
> | Training time (hrs)  | 4.6 | 8.7 | 23.1 | 25.3 |
> | ACR ($\sigma=0.5$) | 0.525 | 0.720 | 0.717 | 0.752 |
>
> ---
> **Q2. How many training samples with the associated generated noises make $p_f=1$? Isn't it a very strong constraint?**
>
> We get $\hat{p}_f(x,y)=1$ whenever $f(x+\delta_i)=y$ for all $i \in \\{1,2,...,M\\}$. i.e., if our classifier predicts the correct labels for all $M$ noises drawn from Gaussian. In our experiment, we found that it is not a harsh condition: e.g., the portion of training samples with $\hat{p}_f=1$ gradually increases up to $54.0\ \\%$ on CIFAR-10 with $\sigma=0.5$. We will add the respective discussion in the final draft.
>
>
> | Epoch  | 0 | 30 | 60 | 90 | 120 | 150 |
> |:--------|:-----:|:-----:|:-----:|:-----:|:-----:|:-----:|
> | $\hat{p}_f=1$ (\%)  |14.2 | 38.6 | 46.6 | 49.0 | 53.0 | 54.0 |
>
> ---
> **Q3. Why $p_0$ is limited to the range $(0,1]$, instead of $(1,\infty)$?**
>
> In our design, $p_0$ represents a target probability value, so it is in (0,1]: we aim to set a requirement on the prediction confidence of the “hard” noises to be at least $p_0$, even they are not primarily optimized in Equation (7). By taking $-\log(\cdot)$ on $p_0$, we get a value that belong to $[0, \infty)$, and this value is used for truncating the cross-entropy losses of hard samples in Equation (7).
>
> ---
> **Q4. Results on ImageNet**
>
> Following your suggestion, we are currently working on ImageNet experiments, and we will include the results in the final draft.
>
> ---
> **Q5. Related works**
>
> Thanks for suggesting some missing references, and we will add it in the final draft. Especially, compared to [Kumar et al., 2020] which also considers the confidence in the context of randomized smoothing, CAT-RS is different from it in the following aspects: (a) CAT-RS uses the confidence information for “training” models, while [Kumar et al., 2020] uses the confidence information for “evaluating” models. (b) [Kumar et al., 2020] uses the confidence of trained models to provide how much one can be confident for the output of models. We will include the other mentioned papers related to general randomized smoothing in the final draft.
>
>
> [Kumar et al., 2020] Certifying confidence via randomized smoothing, NeurIPS 2020.

---

### Official Review · Reviewer_5emU · 2021-11-03

**Correctness:** 3
**Technical Novelty And Significance:** 3
**Empirical Novelty And Significance:** 3
**Recommendation:** 5
**Confidence:** 4

**Main Review:**

Major comments:
- The paper proposes a pretty novel method to boost the training for certified robustness. The experiments look descent to me although the hyperparameter selection seems random to me. I wonder if a cross-validation is performed.
- The intuition behind the loss of high-confidence samples is that the noise samples provided is limited, which cannot guarantee the real $p_f(x,y)$ to be close enough to 1. This is especially important especially for ACR as increasing $p_f(x,y)$ can increase the certified radius more when $p_f(x,y)$ is closer to 1. However, the method used by the authors seems a little arbitrary to me. They find the worst noises within the $\ell_2$-ball of sampled noise by PGD, and somehow normalize the worst noise. Unlike SmoothAdv or simply increasing the number of samples, there is no clear reason why this should work.
- The intuition behind low-confidence part is also natural, basically giving ways to high-confidence part. I think the authors have done that part pretty well.

Minor comments:
- $K$ is used as the number of classes and the binomial random variable.
- Sec 3.1, para 2, $p$ is not known here, so I think the $p$ in binomial distribution should be the empirical estimation $\hat p$.
- Sec 3.2, define $\delta^*$ and how it is normalized.
- Why is the binomial distribution necessary? You can probably just set $K$ to be the $pM$.
- In Table 3, row (b), do you include the masking condition to $L^{high}$? If not, what is the performance if the condition is applied?

**Summary Of The Paper:**

This paper proposed new loss functions during the training of classifiers for certified robustness via randomized smoothing. The new loss function treat samples with different confidence level differently. The main idea is to prioritize samples with high confidence because it provides more additional certified radius when its confidence grows.

**Summary Of The Review:**

My current assessment of this paper is slightly below the threshold because I am not very satisfied with the high-confidence part. I would like to hear from the authors about the mathematical intuition behind the current loss.

---

> ### Author Response · Authors · 2021-11-23
> **Response to Reviewer 5emU**
>
> We sincerely appreciate your thoughtful comments, efforts, and time. We respond to each of your questions and concerns one-by-one in what follows. All the responses will be carefully incorporated in the final draft.
>
> ---
> **Q1. The hyperparameter selection seems random.**
>
> In our experiments, we did not put much efforts on finding the optimal hyperparameters: e.g., (a) we considered only a fixed hyperparameter configuration for Table 1 (MNIST), and (b) even for Table 2 (CIFAR-10), we fixed all but $\lambda \in \\{0.5, 1.0, 2.0\\}$ and $p_0 \in \\{\frac{1}{20}, \frac{1}{10}, \frac{1}{5}\\}$ those affects ACR under the trade-off between accuracy and robustness per $\sigma$. Here, in Table 2, we report the results from best-performing $(\lambda, p_0)$ in terms of ACR, to compare the best achievable ACRs according to the best reported values in each paper of the baselines, as most of the baseline methods also report results across different hyperparameter choices per $\sigma$.
>
>
> ---
> **Q2. Design choice of high-confidence loss**
>
> In order to achieve a high certified radius for a sample $x$, it is important to maximize the count of correct classifications of $f(x+\delta)$ over $n$ samples of i.i.d. Gaussian noise $\delta$ (e.g., $n=100000$ in the practical scenario of CERTIFY [Cohen et al., 2019]), rather than just minimizing the average loss, especially when these losses are already close to 0 for many $\delta$’s. The goal of our high-confidence loss is to find a noise sample that (a) has a high cross-entropy loss, i.e., those are more likely to be incorrect, while (b) keeping the likelihood of the original i.i.d. Gaussian distribution, so that it is likely to be sampled as $\delta$.
>
> [Cohen et al., 2019] Certified adversarial robustness via randomized smoothing, ICML 2019.
>
>
> ---
> **Q3. In Table 3, row (b), do you include the masking condition?**
>
> We do not include the masking condition for the ablation in Table 3-(b), the “$L^{\tt high}$ only”, as the condition in Equation (10) can be valid only with minimizing $L^{\tt low}$: the condition depends on the running confidence $\hat{p}_f(x,y)$, which is optimized through minimizing $L^{\tt low}$. We will clarify this point in the final draft.
>
>
> ---
> **Q4. Minor comments**
>
> Many thanks for the careful reading, and making constructive suggestions to improve the clarity of our manuscript. We will gratefully incorporate all your minor or editorial comments in the final draft.

---

> > ### Comment · Reviewer_5emU · 2021-11-29
> > **Response**
> >
> > I would like to thank the authors for their responses. Some of my concerns have been clarified, but I still have some reservations:
> >
> > - For the hyperparameter choosing, it is absolutely fine to choose different hyperparameters for different $\sigma$, but they should not be directly chosen based on performance on the test dataset. Choosing from a separate validation set is needed to prevent overfitting.
> >
> > - I understand the general intuition for high-confidence loss, but I am confused about the details.  How do the authors decide to find the worst noises within the $\ell_2$-ball (of radius 1) and then somehow normalize them? Why the radius is not even related to $\sigma$? How exactly is the likelihood kept high by normalizing? Comparison for different implementations is needed to justify the current choice.
> >
> > Therefore, I think this paper needs more carefully-designed experiments before it is ready for publication.

---

### Official Review · Reviewer_girT · 2021-11-03

**Correctness:** 3
**Technical Novelty And Significance:** 2
**Empirical Novelty And Significance:** 2
**Recommendation:** 5
**Confidence:** 3

**Main Review:**

While the provided experimental results look promising, the designed loss function (especially for the low-confidence loss) seems to be very complicated in my perspective and the explanation in Section 3 are not clear enough to support such design. In particular, I have the following questions for authors to provide more clarifications:

1. The proposed CAT-RS loss relies on the prediction confidence of a classifier f for selecting examples. Is such f trained in advance or iteratively updated over the training process? How do you obtain the final certified smoothed classifier?

2. I do not understand the design of Equation (7) for the low confidence examples. Discussions in Section 3.1 do not give a clear picture on such specific design choice. Can you provide a clarification on this?

3. Follow the previous question, the cold start problem mentioned in Section 3.1 is also confusing to me. Why Equation (6) leads to a cold start problem? Why Equation (7) resolves this issue? The authors use many terms like “the easiest noise” and “harder ones” in the paragraph above Equation (7), which are vague and hard to parse.

4. Figure 1 and Figure 2 are not well-explained. What do you mean by p_f=0.9 in Figure 1? Is it computed for a single example or computed as an averaged score for the whole dataset? What does areas with different colors represent in Figure 2? I do not understand the colored cross mark in Figure 2 as well.

Other comments:

1. The whole procedure for obtaining the final randomized smoothed classifier should be described in addition to the training loss.

2. It would be better to provide experiments on larger dataset such as Image-Net.



**Summary Of The Paper:**

This paper proposes a loss function for training the base classifier for randomized smoothed classifiers. Specifically, the loss distinguishes between training examples with high prediction confidence and those with low confidence. Several empirical tricks are applied to design the loss function to optimize the performance. Experiments on MNIST and CIFAR-10 show that the proposed training method is superior to existing state-of-the-art randomized smoothed classifiers, especially when the radius r is large.

**Summary Of The Review:**

In summary, the empirical results of the paper show advantages of the proposed training objective in producing better certified smoothed classifiers. However, given the current low clarity of the paper, especially when introducing the design of the proposed loss function, I suggest a weak reject for this work.

---

> ### Author Response · Authors · 2021-11-23
> **Response to Reviewer girT**
>
> We sincerely appreciate your thoughtful comments, efforts, and time. We respond to each of your questions and concerns one-by-one in what follows. All the responses will be carefully incorporated in the final draft.
>
> ---
> **Q1. Do you train the base classifier $f$ in advance, or iteratively update over training?**
>
>
> We do not pre-train $f$ in advance, and the prediction confidences (to compute our proposed loss) can be iteratively updated throughout training $f$. Nevertheless, we previously observed that using a (Gaussian) pre-trained model for the confidence information could slightly improve the certified robustness in our training. However, given the additional training cost from pre-training and its marginal gain, here we only present the case when $f$ is jointly trained with the prediction confidence. We will clarify this point in the final draft.
>
>
> ---
> **Q2. How do you obtain the final smoothed classifier from the base classifier?**
>
> We do not modify the way to obtain the final smoothed classifier from the original randomized smoothing [Cohen et al., 2019], as our focus is to propose a new training method, rather than modifying the certification procedure. More specifically, we obtain the smoothed classifier $\hat{f}$ from $f$ by averaging the output of our trained $f$ under Gaussian noises, as defined in Equation (2). We will clarify this point in the final draft.
>
> [Cohen et al., 2019] Certified adversarial robustness via randomized smoothing, ICML 2019.
>
> ---
> **Q3. Section 3.1: Why "easy"/"hard"?**
>
> Given that an image is perturbed by a (significant) Gaussian noise, the difficulty of classifying that image can vary depending on whether or not the noise blocks some semantic features in the original images. We refer to “hard” and “easy” noise to discriminate between these cases. The key motivation of CAT-RS is to treat the two cases differently in the training objective, given a set of Gaussian noises for each image.
>
> ---
> **Q4. Design of Equation (7) for the low confidence examples**
>
> Our intuition on designing Equation (7) is to prevent the classifier from minimizing the losses from “hard” noises, which may harm the performance: given $M$ i.i.d. Gaussian noises, we only minimize the top $K \sim \mathrm{Bin}(M, \hat{p}_f(x,y))$ easiest noises, i.e., those get lower loss values. This rationalizes the choice of design in Equation (6). The connection between Equation (6) and Equation (7) is summarized in Q5. We will clarify this point in the final draft.
>
> ---
> **Q5. Why does Equation (6) lead to a cold start? Why Equation (7) solves this issue?**
>
> In the early stage of training, it is likely that $K \sim \mathrm{Bin}(M, \hat{p}_f(x,y))$ be zero because the running confidence $\hat{p}_f(x,y)$ is quite lower than the optimal confidence $\overline{p}_f(x,y)$. Then, in Equation (6), the cold start problem occurs because the classifier learns nothing for many $x$’s. We empirically found that clamping $K$ at least 1 can force the classifier to achieve faster convergence by compensating the gap between $\overline{p}_f(x,y)$ and $\hat{p}_f(x,y)$. We will clarify this point in the final draft.
>
>
> ---
> **Q6. Details on Figure 1 and Figure 2**
>
> Thanks for your questions on the clarity of our manuscript, and we will clarify this in the final draft.
>
> Figure 1 plots and compares the certified radius of existing certification methods of smoothed classifiers [Lecuyer et al, 2018; Li et al, 2018; Cohen et al., 2019] in terms of varying $p_f(x, y)$ for a given $x$ and $y$: notice that specifying $p_f(x, y)$ is enough to compute the lower-bound in Equation (3), which is in fact from [Cohen et al., 2019] and the blue plot of Figure 1.
>
> Figure 2 illustrates the overview of our proposed training scheme. The boundary between different colors is the decision boundary of the base classifier, and the light-gray color background is for the true label. The red/blue crosses of (a) are “easy”/”hard” noises we want to use in Equation (7). The red cross of (b) is the “worst” noise we defined in Section 3.2. The arrow from a black cross to the red cross describes the process of finding the “worst” noise in Equation (8).
>
>
> [Cohen et al., 2019] Certified adversarial robustness via randomized smoothing, ICML 2019.
>
> [Lecuyer et al, 2018] Certified robustness to adversarial examples with differential privacy, IEEE S&P 2019.
>
> [Li et al, 2018] Certified adversarial robustness with additive noise, NeurIPS 2019.
>
> ---
> **Q7. Results on ImageNet**
>
> Following your suggestion, we are currently working on ImageNet experiments, and we will include the results in the final draft.

---

### Decision · Program_Chairs · 2022-01-20

**Decision:**

Reject

**Comment:**

This authors seek to improve upon previous work on randomized smoothing for certifiably robust models. They develop loss functions inspired by the notion of distinguishing hard and easy samples while training the base classifier that is randomly smoothed and conduct experiments evaluating their proposed losses on benchmark datasets.

While the reviewers agree that the paper contains interesting ideas, the paper in its current form is unacceptable for publication because:
1) Missing large scale experiments: All prior work on randomized smoothing report results on ImageNet, and this was seen as one of the main advantages of randomized smoothing. Since the authors do not report this, it brings into question the robustness and scalability of improvements obtained.
2) Computational complexity and improvements: The authors' approach has significant computational complexity and the final improvements obtained are marginal. This makes it difficult to justify the use of a more expensive method.